# The Public Health Contribution of Sentiment Analysis of Monkeypox Tweets to Detect Polarities Using the CNN-LSTM Model

**DOI:** 10.3390/vaccines11020312

**Published:** 2023-01-31

**Authors:** Orlando Iparraguirre-Villanueva, Aldo Alvarez-Risco, Jose Luis Herrera Salazar, Saul Beltozar-Clemente, Joselyn Zapata-Paulini, Jaime A. Yáñez, Michael Cabanillas-Carbonell

**Affiliations:** 1Facultad de Ingeniería y Arquitectura, Universidad Autónoma del Perú, Lima 15842, Peru; 2Carrera de Negocios Internacionales Facultad de Ciencias Empresariales y Económicas, Universidad de Lima, Lima 15023, Peru; 3Facultad de Ingeniería, Ciencias y Administración, Universidad Autónoma de Ica, Chincha Alta 11701, Peru; 4Dirección de Cursos Básicos, Universidad Científica del Sur, Lima 15067, Peru; 5Escuela de Posgrado, Universidad Continental, Lima 12000, Peru; 6Vicerrectorado de Investigación, Universidad Norbert Wiener, Lima 15046, Peru; 7Escuela Técnica Superior de Ingenieros de Telecomunicación, Universidad Politécnica de Madrid, 28040 Madrid, Spain

**Keywords:** monkeypox, sentiment, tweets, CNN, LSTM

## Abstract

Monkeypox is a rare disease caused by the monkeypox virus. This disease was considered eradicated in 1980 and was believed to affect rodents and not humans. However, recent years have seen a massive outbreak of monkeypox in humans, setting off worldwide alerts from health agencies. As of September 2022, the number of confirmed cases in Peru had reached 1964. Although most monkeypox patients have been discharged, we cannot neglect the monitoring of the population with respect to the monkeypox virus. Lately, the population has started to express their feelings and opinions through social media, specifically Twitter, as it is the most used social medium and is an ideal space to gather what people think about the monkeypox virus. The information imparted through this medium can be in different formats, such as text, videos, images, audio, etc. The objective of this work is to analyze the positive, negative, and neutral feelings of people who publish their opinions on Twitter with the hashtag #Monkeypox. To find out what people think about this disease, a hybrid-based model architecture built on CNN and LSTM was used to determine the prediction accuracy. The prediction result obtained from the total monkeypox data was 83% accurate. Other performance metrics were also used to evaluate the model, such as specificity, recall level, and F1 score, representing 99%, 85%, and 88%, respectively. The results also showed the polarity of feelings through the CNN-LSTM confusion matrix, where 45.42% of people expressed neither positive nor negative opinions, while 19.45% expressed negative and fearful feelings about this infectious disease. The results of this work contribute to raising public awareness about the monkeypox virus.

## 1. Introduction

The monkeypox virus was first discovered in 1958 in colonies of monkeys being bred for research [1]. It was in the Democratic Republic of Congo that the first human case of monkeypox virus was reported in 1970 [2]. Since then, vaccines against the virus have been developed. In 1980, the monkeypox virus was declared eradicated and vaccination of the population was stopped [3,4]. While it is true that most patients can recover, the mortality rate during a monkeypox outbreak has traditionally ranged from 1% to 10% [5]. Monkeypox is a contagious disease that initially affected countries in Africa; however, it has recently spread to virtually every city in the world. Although the World Health Organization (WHO, Geneva, Switzerland) does not recognize it as a pandemic, some experts believe it should be treated as such [6]. For example, the countries with the highest number of infections from May to 28 August are the United States with 403,937, Spain with 245,977, the United Kingdom with 159,815, Germany with 155,315, and France with 115,187, among others [7], as shown in Figure 1.

Currently, specialists in social data processing are making strenuous efforts to understand the human condition and the psychology of the human condition [8]. Twitter is considered a source of large volumes of data used in the field of behavioral sciences and human psychology, with the purpose of analyzing and knowing the feelings of Internet users [9]. With data processing, it is possible to automatically classify people’s opinions regarding a specific topic into three categories: positive, negative, or neutral. In this work, a hybrid model was used to analyze and quantify the sentiments of Twitter users [10]. The main objective of this work was to analyze the positive, negative, and neutral sentiments posted on Twitter with the hashtag #ViruelaDelMono, through natural language processing (NLP) and machine learning algorithms. Twitter has become an important source of information dissemination through the Internet. The real-time, clear, and accurate communication of current events motivates users to register their opinions and reactions [11]. Twitter is the platform where users share their reactions, feelings, and opinions related to epidemics such as COVID-19, monkeypox, and other epidemics [12,13]. However, to collect all this valuable information provided by Twitter, the use of algorithms based on machine learning (ML) is required, due to a large number of words and contextual phrases that this represents for its processing [14,15]. Sentiment analysis is an NLP technique used to determine positive, negative, and neutral data. It is usually performed on textual data to monitor the opinions of Twitter users [16].

WHO/PAHO, academics, and scientists have published research papers on the epidemiological situation of the monkeypox virus [17,18]. For example, ref. [19] developed an extensible open-source tool that provides a comprehensive approach to assessing the accuracy of monkeypox virus-related claims in order to combat misinformation in digital media. Similarly, ref. [20] presented a regression model based on a CNN capable of handling cells. The algorithm of this model requires a set of training images generating output structures, commonly referred to as proximity patches [21]; these proximity patches present values closer to the center of the cells [22]. On the other hand, Italy has been a very important population focus for studies related to the subject, considering that it was the country with the highest number of people infected by COVID-19 in a given period. In view of this, a model was developed using artificial neural networks to determine potential donors for treatment with COVID-19 convalescent plasma [23]. Along the same lines, two studies developed a model capable of analyzing the themes and sentiments associated with misinformation about the COVID-19 vaccine in social networks [24,25], for which the latent Dirichlet allocation (LDA) machine learning model is used to identify the predominant themes in the misinformation of sentiment-analysis-related research [26]. Classification algorithms are those used for data mining and text mining. That is why the authors of [27] conducted a search on Twitter to collect the main tweets with the word “monkeypox”. The analysis showed that 60% of the tweets posted corresponded to misinformation, followed by information provided by public health officials (which represented 32%) and reports from media or journalists (which represented 8%). To arrive at these results, the information in the tweets was divided into medically correct information and information that was not medically correct. The data published on social networks are data that are worthless without processing [28]. This is why it is important to process these data with the appropriate models. Although there is a set of machine learning models for prediction, it is important to consider the accuracy rates of each one. Therefore, ref. [29] compared 13 pre-trained deep learning models for monkeypox virus detection, using precision, recall, F1 score, and accuracy as measures, with results of 85.44%, 85.47%, 85.40%, and 87.13%, respectively. The present research is organized as follows: Section 1 addresses the research problem, the importance of the study, the purpose, and the objective to be carried out. Section 2 describes the work methodology, the LSTM architecture, the process diagram, and the dataset to be trained. Section 3 presents the results obtained after training. Section 4 presents the discussion. Finally, Section 5 presents the conclusions of the work.

## 2. Material and Methods

This section describes the terminology corresponding to the CNN-LSTM model and the implementation of the work to analyze the positive, negative, and neutral sentiments posted on Twitter Peru with the hashtag #ViruelaDelMono (Spanish for monkeypox).

### 2.1. CNN-LSTM

The hybrid CNN-LSTN model provides more efficient data processing than the stand-alone models. The convolutional layers function as filters of the incoming data, extracting relevant information that is used as input for the new connected convolutional layer. LSTM is a subcategory of the recurrent neural network (RNN); thus, it works very similarly to RNN. This network can add or remove information that it considers relevant for processing. Compared to RNN cells, LSTM cells have an additional input, which is known as a state cell. This cell is the key to the processing of LSTM networks and can function as a conveyor belt to which irrelevant data that we do not want to remain in the network memory can be added or removed. For this work on sentiment categorization based on Twitter data, a hybrid architecture called CNN-LSTM is used to test the model (Figure 2).

In general, traditional RNNs seek to solve problems related to “memory loss”, as this generates unfavorable performance in time series and sequence problems. These models interact cyclically in the hidden layer, thus developing short-term memory, and can extract time series data. However, one of the limitations of RNNs is the loss gradient, which makes it difficult for the model to continue learning. Therefore, LSTMs address this problem by retaining important information in memory cells and removing unimportant information, which results in better performance than a conventional RNN.

The LSTM network has 3 main gates: input, output, and forgetting. This scheme allows the network to maintain a controlled flow of information to determine which data to forget and which to remember, thus learning long-term dependencies. The input gate *i_t_*, together with the second gate, regulates the volume of new memory states *C_t_* at time t. The forgetting gate f determines whether information from the past should be erased or retained in the memory at time *t* − 1, while the output gate *t*
*C** *i* determines which information can be used for the output of the memory cell. Equations (2)–(5) of the activities of an LSTM unit are described below.
(1)$$i_{t}=\sigma (U_{i}x{\;}_{t}+w{\;}_{i}h_{t-1}+b_{i})$$
(2)$$f_{t}=\sigma (U_{g}x{\;}_{t}+\;w_{g}h_{t-1}+b_{g})$$
(3)$$c_{t}=\tan h(U_{c}x{\;}_{t}+w_{c}h_{t-1}+b_{c})$$
(4)$$c_{t}=g_{t}\;¤\;c{\;}_{t-1}+l{\;}_{i}¤c_{t})$$
(5)$$O_{t}=\sigma (U_{o}x{\;}_{t}+w{\;}_{o}h_{t-1}+b_{o})$$
where *x_t_* represents the input, *W** and *U** represent the weight matrices, *b** represents the bias term vectors, σ represents the sigmoid function, and the operator n represents the multiplication.
(6)$$h_{t}=O_{t}⊕\tanh\left(c_{t}\right)$$

In general, the CNN-LSTM model uses the CNN as an encoder for capturing input data characteristics in an LSTM. Figure 1 shows the architecture.

### 2.2. Understanding Data

To develop the case, we used a data set consisting of 84,018 tweets related to the monkeypox virus, which were obtained from Peru in September 2022. As shown in Figure 3, the process involved a series of steps, starting with the import of the dataset, followed by data processing, early cleaning, tokenization, exploratory analysis, classification, and finally the sentiment analysis of opinions. In the following sections, we will take a closer look at each of the points described above in more detail.

### 2.3. Data Cleaning and Processing

Figure 2 describes the general process of sentiment analysis and classification. However, before performing the actual training with tweets, preprocessing is an important step that must be performed first. This starts with the import of the dataset of real tweets. For this work, we managed to extract 84,018 tweets with the hashtag #ViruelaDelMono corresponding to the month of September 2022 in Peru.

Therefore, to carry out the classification and prediction process, it was necessary first to carry out the cleaning process. This consisted in identifying, correcting, or eliminating incomplete, incorrect, or duplicate data that did not contribute to the model. This process was important to enable the model to make more accurate decisions regarding data analysis. This can include everything from checking the accuracy and completeness of the data to filling in missing data and converting the data to a new format. For the cleanup process, there was a set of libraries and tools that Python provides. The most relevant libraries in this work were the following:Pandas: a data analysis library that provides a data structure and tools for cleaning and manipulating data. Panda allows users to eliminate duplicated data, complete missing values, and change column formats, among other cleaning operations.Numpy: a numerical calculation library that provides functions for working with arrays and matrices. Numpy allows users to remove NaN (Not a Number) and perform math on the dataset.OpenRefine: an open-source data-cleansing tool that allows users to quickly and easily clean, transform, and enhance large data sets. In this work, it was used to load data in formats such as CSV, Excel, JSON, and XML. Additionally, the regular expressions technique was released to search for patterns in the data set. This technique helps users find and replace specific patterns of data, that is, only the data that are useful to clean [30].

In this process, stemming was used to replace the words extracted from Twitter with the root word throughout the process. Then, we continued with tokenization, which consists of dividing the text of the sentences into words. Subsequently, a preliminary analysis was performed to understand and analyze the meaning of the words and how often they were used. This work used the programming language Python. For this preliminary analysis process, the DataFrame was used, in which the tweet information was stored. The next step was classification, for which the most commonly used method is the well-known bag of words, which consists of identifying the format for all the words provided by the corpus. Finally, the sentiment analysis was performed. For this analysis, it was necessary to have a dictionary of words in order to associate a feeling with each word.

The collected tweets included 84k tweets contained relevant information related to monkeypox. Within these 84k tweets, there were 90 attributes, so 90 attributes were processed with 84k tweets. Subsequently, lemmatization was applied. This process is based on the fact that the root of a word can be presented in various ways referring to the same thing. For this process, the UDpipe library was used, as shown in Table 1.

The next step was to perform an exploratory analysis of the dataset in order to identify the most commonly used words in the corpus, using the bag of words model. This method is used to present words ignoring their order. In this model, a document is a bag of words, which allows for dictionary-based word modeling, where each bag contains a certain number of words. Figure 4 shows the word cloud, highlighting the most frequent words.

Next, the Syuzhet software was used, which facilitates the extraction of text-derived topics using a variety of opinion dictionaries conveniently packaged for model consumption. This package implements the following dictionaries: Bing, Stanford, Afinn, and NRC. The NRC dictionary was used for this work since it is the only one in the Spanish language. This dictionary has more than 14 thousand words divided into categories of feelings. These categories include positive, negative, and angry emotions, such as anger, fear, disgust, rage, premonition, etc. This dictionary was used to group the words and to understand the categories into which the feelings are divided. Figure 5 shows the feelings that are most frequently used with respect to the monkeypox virus.

Figure 6 shows the words grouped by sentiment, which allowed us to analyze the set of words that are associated with a certain sentiment. For example, the feeling disgust is associated with the following words: enfermedad, contagio, epidemia, secuelas, muerte, and erupción. Along the same lines, the word “fear” is associated with the following words: caso, enfermedad, contagio, riesgo, gobierno, and evitar. In addition, negative sentiment is associated with words such as: virus, caso, enfermedad, riesgo, and evitar. Similarly, the feeling of sadness is associated with the following words: caso, enfermedad, evitar, grave, and emergencia. However, there are also words associated with positive sentiment, such as: ministerio, paciente, vacuna, información, and importante. It is important to look at the positive sentiment. Since there are a certain number of users who do not rate everything negatively, it means that they have confidence in the ministry and in the information provided by reliable media and rate the monkeypox virus vaccine positively.

Up to this stage, we had already cleaned and grouped the data in a document. In the next step, the TM package was used to employ the removeSparseTerms() function, allowing words that do not have many iterations or are not frequent to be removed. Additionally, in this phase, it was necessary to analyze the relationship between words; for this purpose, the function finAssocs() was used. This function calculates the relationship between words and assigns a score that is represented in the line thickness of the association between words. The tweets of the data set in this work were cleaned and organized in such a way that the operation of searching for a relationship between the terms was simple. It involved the use of a vector which could then be used to create the diagram with the ggplot2() library, as shown in Figure 7. For example, the root word virueladelmono is associated with virueladelmono-salud, virueladelmono-contagios, virueladelmono-nuevos, virueladelmono-casos, virueladelmono-país, and virueladelmono-confirmados. This is also followed by the following associations: vacunación-registro, registro-virueladelmono, casos-ministerio, enfermedad-vacunación, vacunación-nacional, riesgo-vacunación, si-riesgo, virus-viruela símica, and entre, among others.

## 3. Results

In this section, the performance of the CNN-LSTM model for sentiment prediction is presented. In this work, the CNN-LSTM model was used to optimize the classification performance and sentiment analysis. This hybrid model uses a set of intermediate layers which can be seen in Figure 1 and Figure 2. For training, the model was applied at different times and with different amounts of data to obtain better model optimization; the training results are shown in Table 2. The results show that the model achieved good results in regard to the four variables: 83% was obtained in accuracy; a 99% rating was obtained in the specificity metric; 85% was obtained in the recovery metric; and, finally, 88% was obtained in the F1 score. Although it is true that 83% was obtained for precision, this is not the best precision metric; however, it can be considered optimal given that the greater the amount of data, the better the performance, and the precision of the model tends to improve.

Figure 8 shows the results after performing sentiment analysis on the data set related to the monkeypox. It can be seen from the results that the sentiments are very unequal, meaning that a large part of the users’ opinions are neither positive nor negative, but neutral. Neutral sentiments represent 45.42% of the data, while negative opinions represent 19.45%. This indicates that the monkeypox virus does not represent generalized negative sentiments. However, 35.13% of opinions regarding the monkeypox virus are positive. Twitter users are probably no longer surprised by a new virus since it closely follows COVID-19. Additionally, users may feel more confident, since we are coming out of a very difficult experience and are now prepared.

The confusion matrix was used to represent the sentiment analysis and classification. Figure 8 demonstrates that the proposed method correctly classifies the sentiments of Twitter users regarding monkeypox.

## 4. Discussion

For this work, a total of 84,018 Twitter feeds were collected to be analyzed and classified with the purpose of determining the feelings of Twitter users regarding the impact of the monkeypox virus. First, people’s reaction was expected to be one of the concerns, especially given that COVID-19 has not yet been eradicated. The increase in the number of monkeypox cases bore some resemblance to the early stages of the COVID-19 pandemic. However, the timely dissemination of information through all media, without neglecting social networks, was crucial. Twitter played an important role as an open platform, allowing the discussion of issues related to public health. Following the performance of the sentiment analysis, as shown in Table 2 and Figure 8, the results indicate that the opinions of Twitter users can be mostly classified as neutral. It can be presumed that users do not have negative or fearful feelings, but that they simply tweeted, pushed by the conjunctural moment of the monkeypox virus. Very similarly, ref. [29] obtained very similar results regarding monkeypox. For example, they found that 60% of the posted tweets related to monkeypox lacked sentiment or corresponded to misinformation, whereas only 32% corresponded to reliable or useful information for processing and classification. The proposed CNN-LSTM model obtained modest results on accuracy, specificity, recall, and F1 score metrics, with averages of 83%, 99%, 85%, and 83%, respectively. Compared to the deep-learning-based approach of [27], better results were obtained in the same metrics of precision, recall, F1 score, and accuracy, with averages of 85.44%, 85.47%, 85.40%, and 87.13%, respectively. This is an indicator of the need to evaluate when choosing pre-trained models for classification and sentiment analysis. ML classification models generally perform better at higher data volumes, which helps to significantly improve accuracy. In comparison with the results obtained in this work, ref. [19] obtained very good results in the accuracy metric (96%), with an open-source automatic learning model based on the BERT algorithm.

### Limitations

The limitations of this study include the rapid evolution of the monkeypox virus and the fact that the analysis and classification were based on Twitter posts by users from Peru, meaning that only tweets in Spanish were considered. Therefore, there is a lack of representativeness relative to other countries, and the results may change over time depending on how the monkeypox virus behaves. Additionally, these results do not necessarily represent the entire population.

## 5. Conclusions

In the field of sentiment analysis, there are various approaches and machine learning models pre-trained to perform these actions. However, it is useful to know which approach and model have the best rates in the measurement metrics. The training, regardless of the model, will depend on the size of the data set; the larger the data volume, the better the model is optimized. In this work, the hybrid CNN-LSTM model was used to perform sentiment analysis and determine the emotional polarity of the content in the tweets. The results obtained in this work are based on user perception regarding the monkeypox virus as positive, negative, or neutral. The model applied in this work, supported by the optimizers, demonstrated high average performance, achieving an average accuracy rate of 83%. According to the other metrics obtained, this is considered a relatively successful approach and suitable for classifying sentiments in tweets related to the monkeypox virus.

In Figure 8, we can see the classification of feelings, based on the data set of 84,018 tweets related to the monkeypox virus. These feelings were classified into the following categories: positive, negative, and neutral, representing 35.13%, 19.45%, and 45.42%, respectively. The results indicate that 45.42% of the users are not afraid of the monkeypox infectious disease. However, 19.45% of the users expressed negative feelings, such as fear, risk, and death, and sought to avoid catching or acquiring the disease by any means of contagion.

The present work has theoretical and practical implications that can be very helpful for future work. Theoretically, we applied a machine learning approach with the hybrid CNN-LSTM model, integrating convolutional networks and a type of recurrent neural network, to optimize prediction results. Practically, this work can be useful for the community as a whole since it helps to raise awareness of the monkeypox virus. Finally, to improve model performance, higher data volume and more robust techniques should be used that can be refined with future research to improve accuracy rates.

## Figures and Tables

**Figure 1 vaccines-11-00312-f001:**
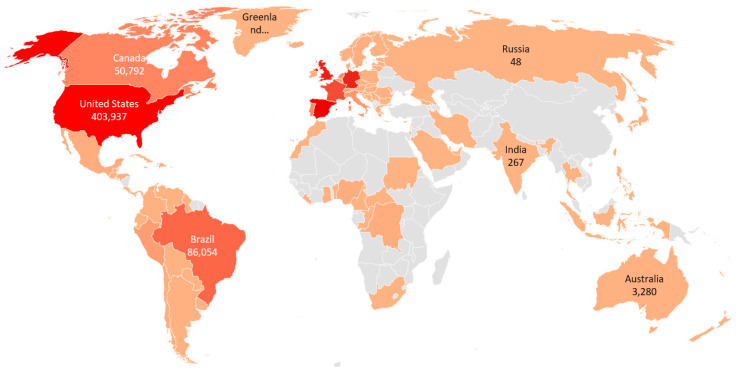
New and cumulative cases of monkeypox worldwide from May through 28 August 2022.

**Figure 2 vaccines-11-00312-f002:**
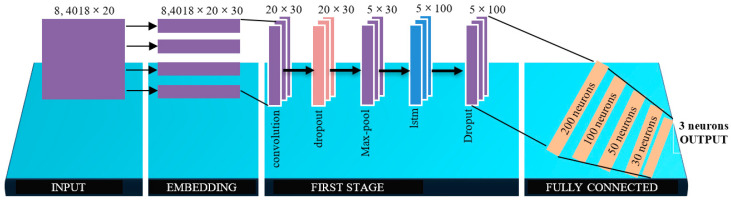
CNN-LSTM architecture for sentiment analysis.

**Figure 3 vaccines-11-00312-f003:**
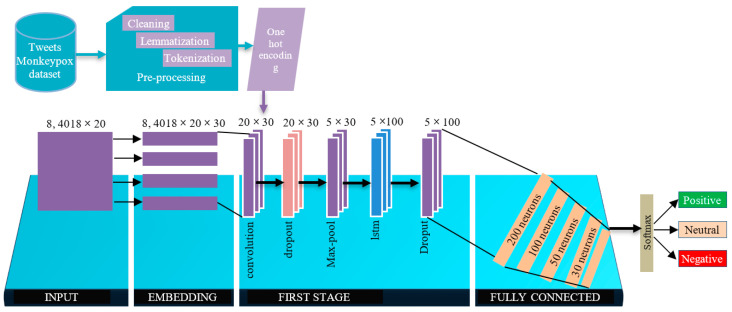
General sentiment classification and analysis process.

**Figure 4 vaccines-11-00312-f004:**
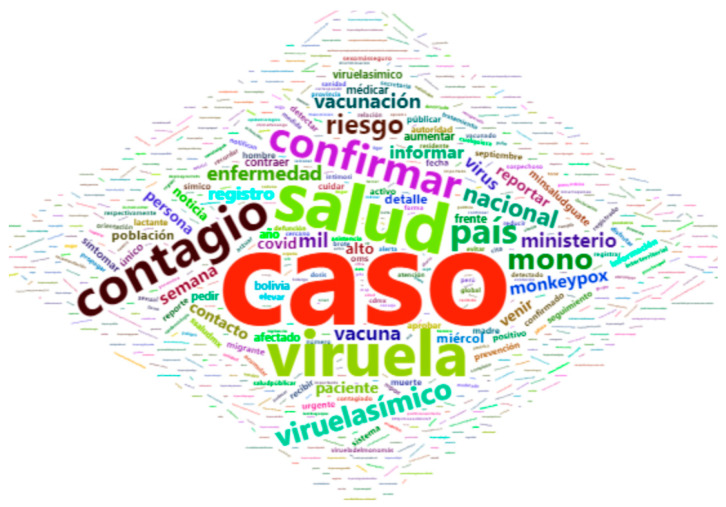
Most frequent words in the corpus.

**Figure 5 vaccines-11-00312-f005:**
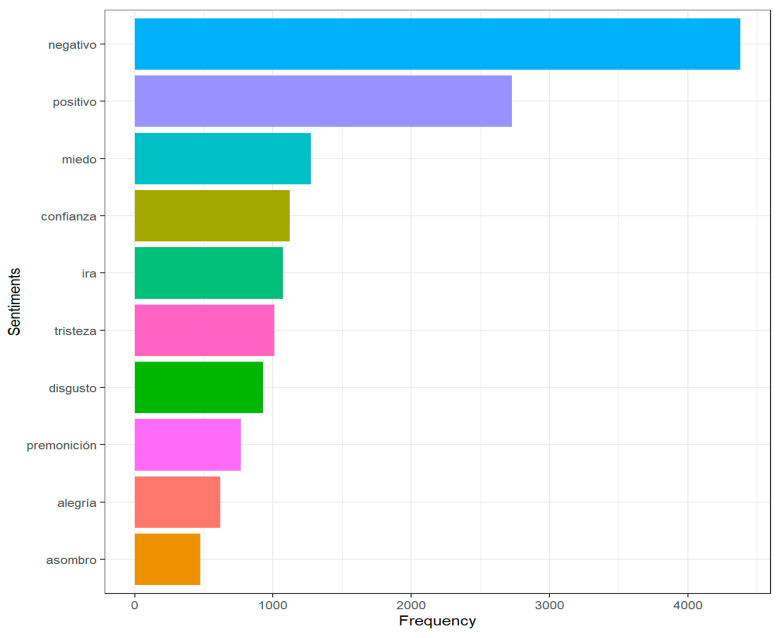
The most frequently used terms.

**Figure 6 vaccines-11-00312-f006:**
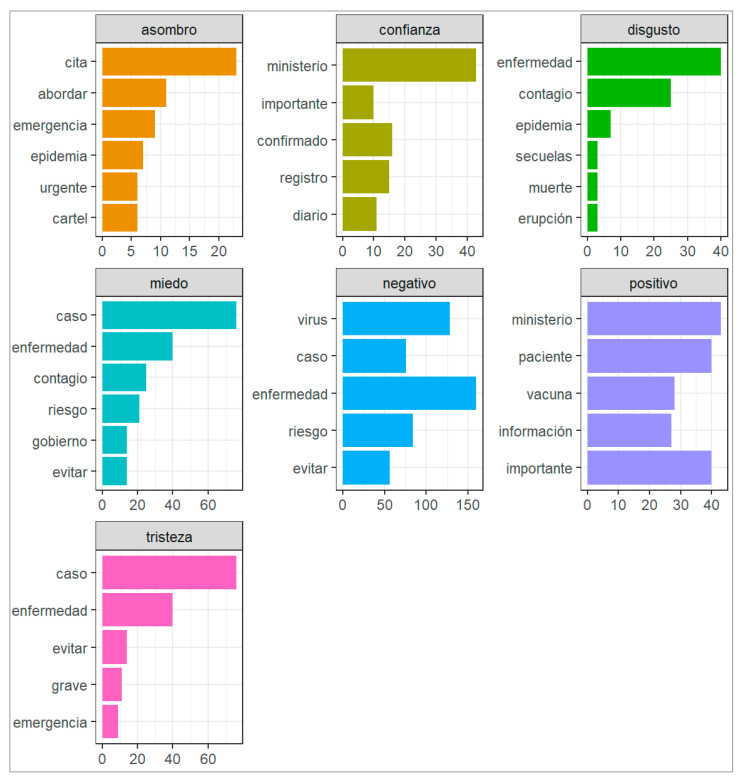
Words grouped by sentiment.

**Figure 7 vaccines-11-00312-f007:**
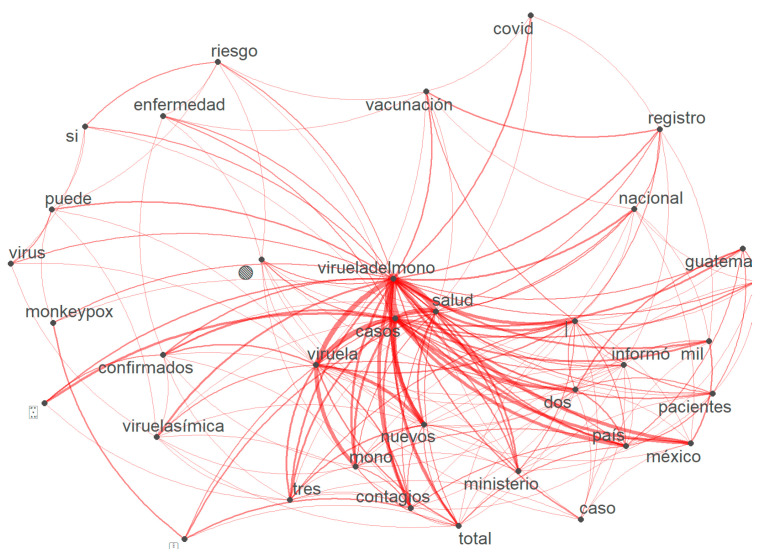
Word association and frequency.

**Figure 8 vaccines-11-00312-f008:**
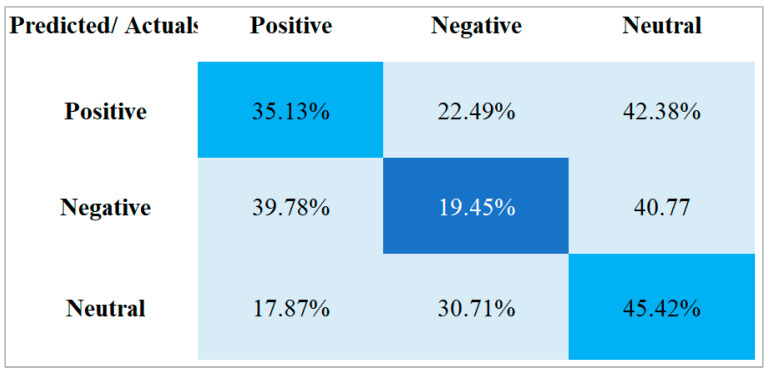
Confusion matrix of CNN-LSTM: Analysis de sentiments.

**Table 1 vaccines-11-00312-t001:** Lemmatization of words.

Sentence_id	Sentence	Token_id	Token	Lema
1	Ministerio	1	Ministerio	Ministerio
1	Salud	1	Salud	Salud
1	Informa	1	Informa	Informar
1	Personas	1	Personas	Persona
1	Dadas	1	Dadas	Dada
1	Alta	1	Alta	Alto
1	Detectado	1	Detectado	Detectado
1	Casos	1	Casos	Caso
1	vienen	1	vienen	Vienir
1	reciebiendo	1	reciebiendo	Recibir
1	médica	1	médica	Médicar
1	Viene	1	Viene	Venir
1	Informa	1	Informa	informar

**Table 2 vaccines-11-00312-t002:** Model performance metrics of CNN-LSTM.

Model	Accuracy	Specificity	Recall	F1
CNN-LSTM1	0.83	0.98	0.89	0.88
CNN-LSTM2	0.83	0.99	0.88	0.89
CNN-LSTM3	0.85	0.98	0.97	0.87
CNN-LSTM4	0.82	0.99	0.76	0.88
CNN-LSTM5	0.79	0.99	0.77	0.89
CNN-LSTM6	0.81	0.98	0.86	0.88
CNN-LSTM7	0.84	0.98	0.85	0.87
CNN-LSTM8	0.86	0.99	0.79	0.88

## Data Availability

Not applicable.

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
