# Peer review of "The Public Health Contribution of Sentiment Analysis of Monkeypox Tweets to Detect Polarities Using the CNN-LSTM Model"

_vaccines, 2023, doi:10.3390/vaccines11020312_

Round 1

Reviewer 1 Report

I was invited to revise the paper entitled "Public health contribution of sentiment analysis of monkeypox tweets to detect polarities using the CNN-LSTM mode". It aimed to evaluate sentiments towards monkeypox pandemic through Twitter. 

The topic is interesting and the methodology is appropriate.

Despite these points, I have several observations:

- The structure of the paper is not appropriate and does not match with a scientific paper. Authors have to present the paper sections as follow: Introduction, Material and Methods, Results and Discussions. Methods were presented both in introduction and methodology sections; results were presented both in Methodology and results section. The paper is confused and not easy to read;

- lines 65-77 should be shifted to methods section;

- Lines 78-83 were useless and should be removed;

- Section 2 should be included in introduction section;

- Discussion is too poor. Firstly, it should be presented in a separate section. In addition it should deeply describe study results, comparing them with previous similar studies. Authors should also declare how to handle these results and why they are important for public health. In addition, Authors  should define what they will do next with these results;

- Strenghts and limitations section should be added in discussions;

- Why did Authors limited their analysis in Perù? does Perù need a particular public health campaign towards MP?

Reviewer 2 Report

The paper presents application of artificial intelligence methodology to the opinions on monkeypox in Peru expressed by Twitter users.

The main objective of the work was to classify the opinions into positive, negative and neutral.

A set of different software was used to prepare 84 th. tweets from September 2022 for analysis (cleaning, removing punctuation and other marks, replacing extracted words with root words, tokenization). Subsequently, a preliminary analysis was performed for understanding the meaning of the words and  to classify them. Then, words which did not have many iterations or were not frequent were removed and relationships between preserved words were analyzed.

Finally, the main method (CNN-LSTM model for sentiment prediction) was applied and evaluated. The results showed polarity of feelings, where 45% Twitter users expressed neither positive nor negative opinions about monkeypox, 19% expressed negative and fearful feelings and 35% - positive feelings.

Despite the possible lack of representativeness of analyzed opinions on monkeypox in Peru, the paper is worth publication – for educational purposes related to the applied methodology. However, possible lack of representativeness should be mentioned in the limitations section.

The methodology of artificial intelligence becomes more and more popular and readers of Vaccines deserve such papers: transparent and relatively easy to understand for non-professionals.

Round 2

Reviewer 1 Report

the paper can now be accepted for publication

Author Response

We thank the reviewer for the positive review of our manuscript.